# Position: Quantum Kernel Machines Should Move Beyond Scalar-Valued Kernels to Realize Their Potential

**Hachem Kadri** [1]  **Joachim Tomasi** [1 2]  **Yuka Hashimoto** [3 4]  **Sandrine Anthoine** [2]

## Abstract

Quantum kernels are reproducing kernel functions built using quantum-mechanical principles and have emerged as a centerpiece of quantum machine learning. The initial enthusiasm for quantum kernel machines has been tempered by recent studies suggesting that quantum kernels could not offer significant computational or statistical advantages when learning from classical data. However, most of the research in this area has been devoted to scalar-valued kernels in standard classification or regression settings for which classical kernel methods are efficient and effective, leaving very little room for improvement with quantum kernels. In this position paper, we argue that progress in this field requires moving beyond scalar-valued kernels toward more expressive kernel frameworks. Scalar-valued kernels lack the degrees of freedom necessary to fully exploit intrinsically quantum resources such as entanglement and are not rich enough to deal with complex learning tasks where classical learning methods struggle. Building on recent advances in operator-valued kernel learning and $C^*$-algebraic kernel representations, we propose a roadmap for designing quantum kernels capable of leveraging entanglement and non-commutative structures to tackle complex structured prediction problems. To support this viewpoint, we present an initial proof-of-concept illustrating how quantum operator-valued kernel formulations can reveal structural dependencies that remain difficult to access for scalar-valued kernel methods. This shift in focus could open a pathway toward a new generation of quantum kernel machines and a more faithful exploration of their potential advantages.

## 1. Introduction

Quantum machine learning (QML) is an emerging field of research at the intersection between machine learning and quantum computing with the goal of using quantum computing paradigms and technologies to improve the speed and performance of learning algorithms (see Figure 1) (Biamonte et al., 2017; Du et al., 2025). Since the seminal works by Havlíček et al. (2019) and Schuld & Killoran (2019), quantum kernel machines have generated a lot of enthusiasm in this field, especially for exploring the applications of noisy intermediate-scale quantum computers to machine learning (Mengoni & Di Pierro, 2019; Blank et al., 2020; Wang et al., 2021; Heyraud et al., 2022). This enthusiasm has been tempered by recent studies that have suggested that quantum kernels could not offer speed-ups when learning on classical data (Huang et al., 2021; Kübler et al., 2021; Jerbi et al., 2023). However, these works have dealt with scalar-valued classification and regression tasks for which classical kernel methods are efficient and effective, leaving very little room for improvement with quantum kernels. We strongly believe that we have to focus on vector-valued, matrix-valued, or structured-output learning tasks where classical kernel methods face clear limitations in order to reveal the full potential of quantum kernels. This is the lens through which we will explore quantum kernel machines, relying on recent advances in classical ML. We present here our point of view on this ongoing field of research and argue that **quantum kernel research should move beyond the restrictive setting of scalar-valued kernels and shift its attention toward more expressive kernel frameworks to realize the full potential of quantum kernels**.

Operator-valued kernels (OVKs) generalize standard kernel functions and offer the possibility of tackling various ML problems ranging from multitask learning to multiview learning and differential equations modeling (Micchelli & Pontil, 2005; Evgeniou et al., 2005; Alvarez et al., 2012; Lim et al., 2015; Kadri et al., 2016; Huusari et al., 2018; Stepaniants, 2023). In this paper, we advocate for the adoption of operator-valued kernels for QML. The operator-valued kernel framework is flexible and gives rise to more expressive quantum feature spaces. We also discuss $C^*$-algebra-valued kernels as a generalization of OVKs. This could open up a new way to apply mathematical theories to QML for struc-

[1]Aix-Marseille University, CNRS, LIS, Marseille, France [2]Aix-Marseille University, CNRS, I2M, Marseille, France [3]NTT, Inc., Tokyo, Japan [4]RIKEN AIP, Tokyo, Japan. Correspondence to: Hachem Kadri <hachem.kadri@univ-amu.fr>.

*Proceedings of the 43rd International Conference on Machine Learning*, Seoul, South Korea. PMLR 306, 2026. Copyright 2026 by the author(s).

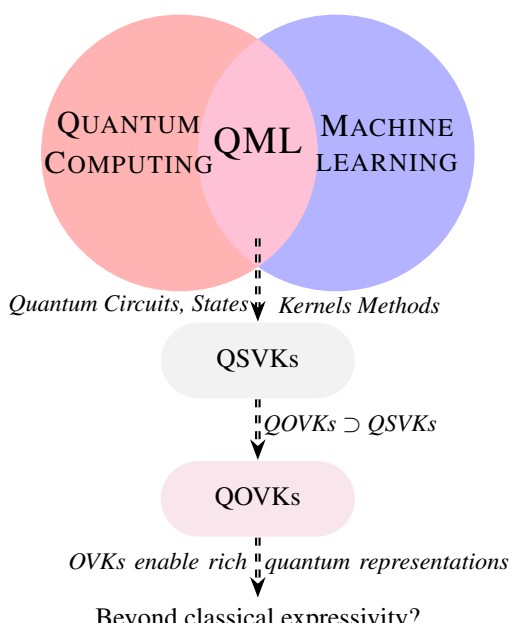

*Figure 1.* Quantum machine learning (QML) is a recent field of research at the intersection between quantum computing (QC) and machine learning (ML). The interaction is two-sided: quantum-enhanced ML (from QC to ML), and ML-based quantum computing (from ML to QC). Most of the interest has concentrated on the use of quantum computing paradigms to improve machine learning algorithms. Quantum scalar-valued kernels (QSVKs) have generated a great deal of interest in the field of QML due to their natural alignment with the kernel trick and their compatibility with hybrid quantum-classical architectures. However, recent findings suggest that their expressive power may be limited in classical data regimes. Quantum operator-valued kernels (QOVKs) offer a more general and expressive framework, potentially unlocking richer hypothesis spaces that are inaccessible to classical or quantum SVKs. QOVKs generalize QSVKs and provide opportunities to go beyond classical expressivity.

tured prediction tasks. Structured output learning is the task of learning a mapping between objects of different nature that each can be characterized by complex data structures such as curves, trees and graphs (Tsochantaridis et al., 2005; Geurts et al., 2006; Kadri et al., 2013; Brouard et al., 2016). Quantum structured prediction should extend the application scope of quantum kernels. Modeling structured dependencies across outputs and complex input-output couplings involves non-separable interactions that scalar-valued kernels are not designed to capture. Operator-valued kernels provide a natural framework for learning such dependencies. This motivates the investigation of their quantum counterpart. To substantiate our viewpoint, we outline a possible quantum implementation of OVKs and provide a proof-of-concept demonstration on quantum channel estimation, framed as a matrix-valued prediction problem. This highlights how operator-valued formulations can outperform traditional scalar-valued kernels.

**Notation** The so-called 'bra-ket' notation is used to describe the state of a quantum system. A column vector $\psi$ is represented as 'ket' $|\psi\rangle$. The conjugate transpose (Hermitian transpose) of a ket, a row vector, is denoted by 'bra' $\langle\psi| := |\psi\rangle^{\dagger}$, where $\dagger$ denotes the conjugate transpose. The inner product of two vectors $|\psi_1\rangle$ and $|\psi_2\rangle$ can be written in bra-ket notation as $\langle\psi_1|\psi_2\rangle$, and their tensor product, can be expressed as $|\psi_1\rangle |\psi_2\rangle$, i.e., $|\psi_1\rangle \otimes |\psi_2\rangle$.

## 2. Kernels from Classical to Quantum

Machine learning is the branch of artificial intelligence that seeks to develop computer systems which detect patterns in data in order to improve their performance automatically with experience (Jordan & Mitchell, 2015). The widespreading development of acquisition tools together with increasing storage capacities has had us witness an explosion in the amount of available data, as well as the urge to develop methods to handle them properly. In this context, it is crucial to design new large-scale machine learning systems that are able to deal with big data.

Quantum machine learning is a relatively recent field of research (Biamonte et al., 2017; Ciliberto et al., 2018; Dunjko & Briegel, 2018). This research field is largely driven by the desire to develop artificial intelligence that uses quantum technologies to improve the speed and performance of learning algorithms. There is also interest in investigating the use of machine learning for tackling quantum computing and quantum information problems. The field is evolving rapidly, but many open questions remain and should be addressed to better understand how quantum computers may outperform classical computers on machine learning tasks.

This paper identifies potential effective interactions between the fields of quantum computing and kernel machines and lays the ground towards a deeper understanding of what kernel-based learning looks like in a quantum world.

### 2.1. Classical Kernel Methods

The research work described in this paper belongs to a large class of learning algorithms, the so-called kernel methods. Since they were introduced by Boser et al. (1992) as a way to construct a nonlinear extension of Support Vector Machines, these methods became very popular. Kernel methods exploit training data through implicit definition of a similarity between data points that can be expressed as a dot product in a feature space, namely reproducing kernel Hilbert space (RKHS). The idea is to transform the data via a feature map $\phi$ into a suitable feature space in which linear learning algorithms could be applied. The inner product between features can be computed using the kernel function; this is the well-known kernel trick, i.e., $k(x,y) = \langle\phi(x), \phi(y)\rangle$ (see Figure 2). It should be pointed out that the notion of kernels

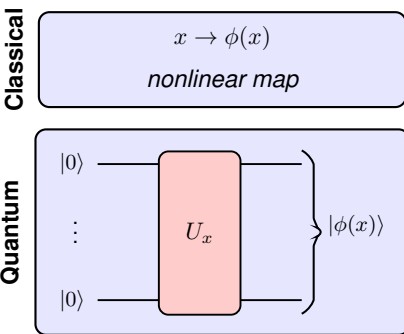

*Figure 2.* Kernel feature map. (top) In the classical setting, a data point $x$ is mapped to a high-dimensional space via an implicit nonlinear feature map $\phi$. The mapping $\phi$ may be unknown but the inner product between two data points, $x$ and $y$, mapped by $\phi$ is equal to the kernel function evaluated at $x$ and $y$, i.e., $\langle \phi(x), \phi(y) \rangle = k(x, y)$. (bottom) In the quantum setting, the feature map is known explicitly. Encoding a data point $x$ into a quantum state $|\phi(x)\rangle$ using a unitary matrix (quantum gate) $U_x$ defines a quantum feature map. A quantum kernel is then defined as the fidelity between two data-encoding quantum feature states, i.e., $k(x, y) = |\langle \phi(x)|\phi(y) \rangle|^2$.

as dot products in Hilbert spaces was first brought to the field of machine learning by Aizerman et al. (1964), while the theoretical foundation of reproducing kernels and their Hilbert spaces dates back to at least Aronszajn (1950). Kernel methods became a mature field able to address many problems in machine learning and statistical data analysis (Hofmann et al., 2008). The tremendous achievements in the field show that these methods provide elegant and powerful learning algorithms for analyzing nonlinear features and processing complex data structures, and offer a comprehensive suite of mathematically well-founded nonparametric modeling techniques for a wide range of learning problems. One major limitation of kernel methods is their high computational cost when the number of training examples is large. This motivates the study of the impact of quantum computation in their computational capabilities. This is of importance since it can give rise to novel and effective strategies to scale up kernel methods for large-scale problems.

## 2.2. Quantum Kernels

A major difference between quantum computing and its classical counterpart is that information is carried by qubits (Nielsen & Chuang, 2010; de Wolf, 2019). Unlike bits which have only two possible states, 0 and 1, qubits can exist in those and any combination of them. More formally, a qubit is an element of a 2-dimensional Hilbert space $\mathcal{H}$, a complex inner product space that is also a complete metric space with respect to the distance function induced by the inner product. An arbitrary qubit $|\psi\rangle$ may be written as $a_0 |\psi_0\rangle + a_1 |\psi_1\rangle$, where $|\psi_0\rangle$ and $|\psi_1\rangle$ form a complete orthonormal basis of $\mathcal{H}$. Multiple qubit states can be ob-

tained from the tensor product of qubit states. A quantum machine learning algorithm needs data in the form of quantum states. Therefore, classical data should be first encoded into quantum states, i.e., the transformation of a classical data $x$ to a quantum state $|\phi(x)\rangle$. Most of the interest in quantum kernels comes from the observation that encoding classical data into a quantum computer defines an explicit feature representation of the data (see Figure 2). Moreover, all operations that can be performed on quantum feature states are linear. This draws a parallel with classical kernel machines (Havlíček et al., 2019; Schuld & Killoran, 2019). Using the kernel trick, a quantum kernel is then defined as the fidelity between two data-encoding feature states, i.e., $k(x, y) = |\langle \phi(x)|\phi(y) \rangle|^2$. Different data-encoding strategies and their quantum kernels have been proposed in the literature (see, e.g., Schuld & Killoran 2019).

## 2.3. The Good, the Bad and the Ugly

Quantum kernels have generated a lot of interest in the field of QML (Blank et al., 2020; Coyle et al., 2020; Kusumoto et al., 2021; Glick et al., 2024). The analogy between quantum data encoding and kernel feature representation provides a conceptual framework for understanding and analyzing quantum machine learning algorithms. This sheds light on the synergies between kernel machines and quantum computing and leads to more interaction between the two fields, whether theoretical or practical. Also, quantum kernels provide a scheme with which to realize hybrid quantum-classical learning. A quantum computer can be used to create feature representations and compute quantum kernels which are then fed into classical learning algorithms (Liu et al., 2021). This makes them suitable for the noisy intermediate-scale quantum (NISQ) era (Preskill, 2018), where quantum computation has to be performed with limited quantum resources.

From another point of view, this analogy is too narrow to support a quantum advantage for machine learning. Some recent studies have argued that supervised quantum machine learning models are kernel methods (Schuld, 2021) or showed that random Fourier features, a widely known method for kernel function approximation, are able to classically approximate variational quantum machine learning (Landman et al., 2023). This leads to the question of whether quantum advantage is the right goal for quantum machine learning (Schuld & Killoran, 2022). Moreover, most of quantum data encoding strategies result in kernel functions that are already known and/or efficiently computable by a classical computer. Whether there are interesting kernel functions that can be computed via quantum states and are classically intractable is still an open question.

More problematic is the generalization ability of quantum ML methods based on kernel functions. Recent studies ana-

lyzed generalization error bounds for learning with quantum kernels and the results appear to be negative (Huang et al., 2021; Kübler et al., 2021). The expressive power of quantum models may hinder generalization. Finding suitable quantum kernels is not easy because the kernel evaluation might require exponentially many measurements. In other words, when using a large number of qubits, the kernel matrix (i.e., the matrix obtained by evaluating the kernel function on all pairs of data points) gets close to the identity matrix, resulting in overfitting and poor generalization performance (Suzuki et al., 2024).

### 2.4. To Be or Not to Be

Most of the previous studies have focused on supervised learning of scalar-valued functions in the context of standard classification or regression. Classical kernel machines in this setting are well-established models and have been extensively studied in the last three decades. Efficient kernel approximations with randomization techniques have been proposed to reduce their computation and storage requirements while performing quite well in various applications (Liu et al., 2022). Moreover, deep learning, which finds its root in the field of neural networks, has enabled tremendous progress for learning on various types of datasets, such as image, language or audio datasets, and achieved impressive performance on classification and regression tasks (LeCun et al., 2015). All these do not leave much space for improvement with quantum kernels in the context of standard supervised learning.

In this paper, we propose moving beyond current quantum kernel models toward operator-valued kernels, enabling a shift from simple kernel similarities to a new generation of quantum kernel machines capable of tackling challenging learning tasks. To realize this shift, we advocate for a research agenda centered on three pillars: (i) the development of quantum operator-valued kernels (QOVKs) that leverage entanglement as a computational resource, (ii) the adoption of $C^*$-algebraic frameworks to exploit non-commutative data structures, and (iii) a strategic focus on quantum structured prediction. This roadmap integrates advances in statistical kernel theory and structured output learning with quantum computing and information.

## 3. Moving Beyond Scalar-Valued to Operator-Valued Quantum Kernels

**Classical OVKs**  Operator-valued kernels (OVKs) appropriately generalize the well-known notion of reproducing kernels and provide a means for extending the theory of reproducing kernel Hilbert spaces from scalar- to vector-valued functions. They were introduced as a machine learning tool by Micchelli & Pontil (2005) and have since been investigated for use in various machine learning tasks, in-

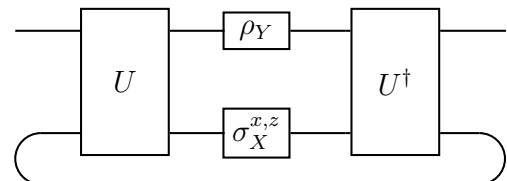

*Figure 3.* Diagram representation of the quantum operator-valued kernel (1). Input data are embedded into a feature matrix $\sigma_X^{x,z}$. We dilate the input system by tensorizing $\sigma_X^{x,z}$ with an output density matrix $\rho_Y$ to form a larger composite system that enables interactions between input and outputs. A (unitary) interaction is then applied via $U$, resulting in the evolution of the composite input-output system. By tracing out the input system, we obtain the final state on outputs, which defines the value of the kernel function ($K(x, z)$ is a matrix acting on outputs).

cluding multitask learning (Evgeniou et al., 2005), functional or operator regression (Kadri et al., 2016), multiview learning (Huusari et al., 2018), and PDE (partial differential equations) learning (Stepaniants, 2023). Despite this progress, the current status of the field of operator-valued kernels suggests further explorations to shed fresh light on old questions, frame new ones and potentially offer new alternatives to existing kernel machines. For more details on (classical) operator-valued kernels and their associated reproducing Hilbert spaces, see Appendix A.

**From classical to quantum OVKs**  A major limitation of operator-valued kernels is their high computational expense. In contrast to the scalar-valued case, the kernel matrix associated to a reproducing operator-valued kernel is a block matrix of dimension $np \times np$, where $n$ is the number of data samples and $p$ the dimension of the output space. Manipulating and inverting matrices of this size become particularly problematic when dealing with large $n$ and $p$. Moreover, questions on how to design operator-valued kernels, what sort of interactions they should learn and quantify and how they should learn them from data are still open. Quantum-based kernel machines represent a promising approach for addressing these challenges. The kernel matrix plays a central role in solving kernel learning problems. For many ML tasks, computing the solution involves matrix-vector operations and solving optimization problems. Quantum linear system solvers (Morales et al., 2026) and quantum optimization algorithms (Abbas et al., 2024) hold potential for addressing these tasks more efficiently than their classical counterparts. The kernel matrix in the operator-valued setting can be significantly larger than its scalar-valued counterpart. This suggests that quantum algorithms could have a more substantial impact in the operator-valued setting, although the implementation of quantum solvers on current quantum devices still faces many challenges.

Quantum operator-valued kernels (QOVKs) have not been investigated yet. Building on the framework introduced

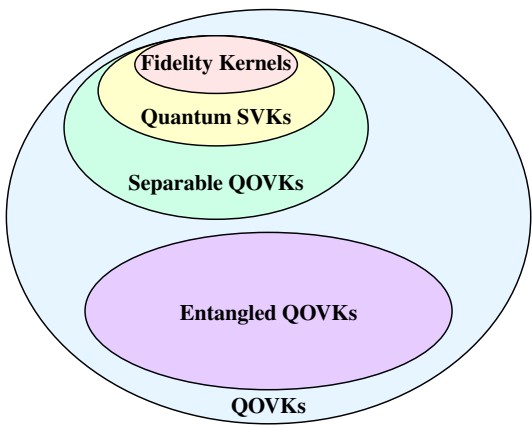

*Figure 4.* Illustration of inclusions among quantum kernel classes considered in this paper. Entangled quantum operator-valued kernels (QOVKs) are not separable, i.e., dependencies between input and output variables cannot be considered separately. The class of separable QOVKs coincides with the class of quantum scalar-valued kernels (QSVKs) when the output dimension is equal to one. If in addition the input feature matrix takes the form of a product of two pure density matrices, the separable QOVKs class becomes the fidelity kernel class.

by Huusari & Kadri (2021), we extend the notion of operator-valued kernels to the quantum domain, introducing the concept of *entangled quantum operator-valued kernel* (QOVK).

**Definition 3.1.** (Entangled QOVK)

An entangled quantum operator-valued kernel $K : \mathbb{C}^d \times \mathbb{C}^d \to \mathbb{C}^{p \times p}$ is defined, $\forall x, z \in \mathbb{C}^d$, $p > 1$, as

$$K(x, z) = \mathrm{Tr}_X \left[ U_{YX} \left( \rho_Y \otimes \sigma_X^{x,z} \right) U_{YX}^\dagger \right], \quad (1)$$

where $U_{YX} \in \mathbb{C}^{pm \times pm}$ is a non-separable unitary matrix (i.e., a unitary matrix that cannot be written as $U_{YX} = A_Y \otimes B_X$, with $A \in \mathbb{C}^{p \times p}$ and $B \in \mathbb{C}^{m \times m}$), $p$ is the dimension of output data, and $m$ is the dimension of input features. In addition, $\rho_Y \in \mathbb{C}^{p \times p}$ is a density matrix on the output space, $\sigma_X^{x,z} \in \mathbb{C}^{m \times m}$ is a feature matrix extracted from inputs $x$ and $z$, and $\mathrm{Tr}_X$ is the partial trace on $X$.

A diagram representation of entangled quantum operator-valued kernel is given in Figure 3.

*Remark* 3.2. It is worth noting that when $U_{YX}$ is separable and equals to $I \otimes B_X$ the kernel $K(x, z)$ in (1) simplifies to a *separable quantum operator-valued kernel* computed using the scalar-valued quantum kernel $\mathrm{Tr}(\sigma_X^{x,z})$, i.e., $K(x, z) = \mathrm{Tr}(\sigma_X^{x,z}) \rho_Y$ (recall that $B_X$ is a unitary matrix).

*Remark* 3.3. When the output dimension $p$ is equal to one, the class of separable QOVK coincides with the class of *quantum scalar-valued kernels*. Moreover, if $\sigma_X^{x,z} = \rho_X^x \rho_X^z$, where $\rho_X^x$ and $\rho_X^z$ are pure density matrices (i.e., $\rho_X^x = |\phi(x)\rangle \langle\phi(x)|$ and $\rho_X^z = |\phi(z)\rangle \langle\phi(z)|$), we recover the class of *fidelity kernels* illustrated in Figure 2.

An illustration of inclusions among the quantum kernel classes discussed above is provided in Figure 4. It is easy to see that scalar-valued kernels can be recovered from operator-valued kernels by considering a separable kernel built using a scalar-valued quantum kernel on inputs and a density matrix on outputs, i.e., $K(x, z) = k(x, z)\rho_Y$. This results in a kernel matrix $G$ of the form $g \otimes \rho_Y$, where $g$ is the scalar-valued kernel matrix. If we restrict ourselves to this class of kernels, QOVKs will suffer from the same limitations as QSVKs. The crucial question is whether alternative classes of OVKs might provide better mechanisms for addressing these limitations.

**QSVK vs QOVK**   The key distinction between QSVKs and entangled QOVKs lies in the way interactions between inputs and outputs are represented. SVKs quantify similarity using a single scalar quantity and therefore cannot directly model structured dependencies among outputs. Even separable QOVKs remain limited because they factorize input similarity and output interactions into independent components. In contrast, entangled QOVKs allow joint, non-factorizable representations of input-output interactions. The use of non-separable unitary operators enables correlations between outputs to depend explicitly on the input representation itself. This is particularly relevant in structured prediction where output relationships are context-dependent. For example, in graph prediction (Geurts et al., 2006) or multi-task learning (Evgeniou et al., 2005), dependencies between predicted outputs may vary according to the input structure, which cannot naturally be represented by separable SVKs.

**Are QOVKs a 'magic bullet' for quantum kernel machines?**   At present, there is no definitive answer. One might reasonably suspect that QOVKs inherit some of the limitations of quantum scalar-valued kernels, which could raise doubts about their usefulness. However, as operator-valued kernels strictly generalize scalar-valued ones, they offer additional structure and degrees of freedom whose impact remains largely unexplored. Determining whether, and to what extent, QOVKs overcome the limitations of QSVKs therefore calls for a systematic investigation.

The scalar-valued kernel framework may be too restrictive to effectively demonstrate the potential advantage of quantum kernel machines over classical kernel methods. In other words, the SVK framework lacks sufficient degrees of freedom to effectively address the challenges and exploit the unique capabilities of quantum kernels. QSVKs are a special case of QOVKs which correspond to simple, separable QOVKs. Entanglement can be a key resource for achieving quantum advantage (Jozsa & Linden, 2003; Wang et al., 2024). The framework of operator-valued kernels provides a means to investigate entanglement and incorporate it into the kernel learning process. This allows us to explore richer

and more complex functions that could be learned more efficiently using quantum computation. Moreover, a key advantage of operator-valued kernels is their ability to naturally integrate input data from multiple modalities and targets from multiple tasks. This significantly improves and expands the application potentials of quantum kernels.

Entangled QOVKs, unlike separable kernels, do not have a Kronecker product structure and can be constructed using quantum correlations. This may open the door for the design of quantum kernels which can be implemented quantumly much more efficiently than classically. Entanglement can play a role in speeding-up quantum computation (Jozsa & Linden, 2003) and OVKs offer a framework for identifying how entanglement may contribute to achieving quantum advantage in kernel-based learning. On the other hand, operator-valued kernels naturally incorporate more data structure than scalar-valued kernels. Adding structure could improve generalization performance and is a well known technique for mitigating overfitting when enhancing expressivity. Entanglement can also have an impact on the number of measurements (Nakhl et al., 2024), which could offer new possibilities for the generalization of quantum kernels. Entanglement is not assumed to automatically confer a quantum advantage; rather, the operator-valued framework provides a principled setting for systematically investigating how entanglement affects learning with quantum kernels.

**Computational perspective**   Classically, SVK methods typically require $O(n^3)$ time and $O(n^2)$ memory due to the construction, storage and inversion of an $n \times n$ Gram matrix. In the operator-valued setting, the kernel matrix becomes a block matrix of size $np \times np$. A direct dense implementation therefore scales as $O(n^3 p^3)$ time and $O(n^2 p^2)$ memory. This substantially increases the computational burden, since matrix inversion, eigendecomposition and related optimization routines scale with the full block matrix. This increased cost is also one reason why the operator-valued setting may provide a more relevant regime for investigating quantum computational benefits. For separable OVKs, Kronecker or Sylvester structure can often be exploited to reduce the computational burden. However, this computational simplification comes with a modeling restriction: separability imposes a factorized structure between input similarities and output interactions. This is precisely the restriction that entangled QOVKs are designed to move beyond.

On the quantum side, two classes of routines are particularly relevant. First, quantum linear-system algorithms, starting with HHL (Harrow et al., 2009) and including more recent block-encoding and QSVT-based approaches (Gilyén et al., 2019), can prepare quantum states proportional to the solution of certain linear systems with polylogarithmic dependence on the matrix dimension, under assumptions on state preparation, block-encoding, conditioning and read-

out (Morales et al., 2026). Second, Grover search and amplitude amplification can provide quadratic improvements in search and sampling subroutines (Grover, 1996; Brassard et al., 2002). Moving from SVKs to OVKs changes the relevant matrix dimension from $n$ to $np$. While dense classical routines scale polynomially in this dimension, quantum routines can in principle exhibit more favorable dependence on $n$ and $p$. Therefore, the growth of the operator-valued kernel matrix may create more room for potential quantum improvements than in the scalar-valued setting.

We stress however that this observation alone is insufficient to establish a quantum advantage for QOVKs. Such an advantage would depend on additional assumptions such as the availability of efficient state preparation and quantum access to data. The point is that the operator-valued setting defines a structurally richer and computationally more demanding regime in which the possible role of quantum resources can be investigated more meaningfully.

## 4. Call to Action

To enable the transition from scalar-valued to more expressive operator-valued quantum kernel machines, we propose the following actionable steps.

**Action 1: Quantum *implementation* of OVKs**   Extending quantum data-encoding schemes and providing quantum circuit implementations of OVKs is of great importance to be able to characterize how well these kernels fit into a quantum computer. Quantum states can be represented by density operators, which are positive semi-definite, self-adjoint operators with unit trace. Identifying and exploring synergies between density operator formalism and OVKs is an interesting path to investigate. Furthermore, quantum superposition, a fundamental concept in QC, is the means by which quantum algorithms like Grover's search can outperform classical ones (Grover, 1996). The objective is also to design quantum algorithms based on superposition for learning with OVKs in order to provide non-trivial improvements in terms of not only their computational complexity but also their statistical efficiency (Roget et al., 2022).

**Action 2: Quantum *entangled* OVKs**   Some classes of operator-valued kernels have been proposed in the literature, with separable kernels being one of the most widely used for learning vector-valued functions due to their simplicity and computational efficiency. These kernels are formulated as a product between a kernel function for the input space alone, and a matrix that encodes the interactions among the outputs. However, there are limitations in using separable kernels. They use only one output matrix and one input kernel function and thus cannot capture different kinds of dependencies and correlations, and assume a strong repetitive structure in the operator-valued kernel matrix that models input and

output interactions. As discussed above, *entangled* QOVKs go beyond separable kernels and offer new opportunities for quantum kernel design. This class should be better investigated to shed light on its potential in finding correlations that cannot be described by classical statistics.

**Action 3: A $C^*$-*algebraic* detour** $C^*$-algebras provide a unified framework for an operational formulation of classical and quantum mechanics (Bru & Pedra, 2023). Reproducing kernel Hilbert $C^*$-module (RKHM) is a generalization of reproducing kernel Hilbert space (RKHS) by means of $C^*$-algebra (Hashimoto et al., 2021). Recently, Hashimoto et al. (2023b;a; 2024) have paved the way for supervised learning in RKHMs. This provides a new twist to the state-of-the-art kernel-based learning algorithms and the development of a novel kind of reproducing kernels. Advantages of RKHM over RKHS are that we can make use of: (i) the $C^*$-algebra characterizing the RKHM to construct rich feature representations and explore a larger function space (Hashimoto et al., 2023b), and (ii) the properties of $C^*$-algebras such as operator norm and spectral truncation to achieve better generalization and design kernels that control local and global dependencies of output data on input data (Hashimoto et al., 2023a; 2026).

From the perspective of the connection with quantum, $C^*$-algebra has rich notions related to quantum mechanics. For example, we can represent quantum gates and density operators as elements of a $C^*$-algebra. Thus, we can obtain them as outputs of the kernel machines with RKHMs. While the application of $C^*$-algebra to QML is a promising way to design quantum kernels, we need further investigations to relate theory with practice. It would be interesting to : (i) investigate connections between $C^*$-algebra-valued kernels and quantum information with a particular attention to learning in RKHM quantum systems (Gebhart et al., 2023); (ii) study the impact of quantum computing on the computational complexity of learning in RKHM.

It is worth noting that solving learning problems with such kernels involves tackling optimization problems over noncommutative groups. Noncommutative optimization (Burgdorf et al., 2016; Bürgisser et al., 2019) is a promising direction to explore within this framework since it makes connections with both noncommutative kernels (Belinschi et al., 2023; Hashimoto et al., 2026) and quantum information (Gribling et al., 2018). It will be interesting to investigate whether noncommutative optimization techniques can improve learning with quantum OVKs.

**Action 4: Application to quantum *structured* prediction** OVKs hold promise to expand the application realm of quantum kernels. Many learning problems of practical interest are intrinsically vector-valued, matrix-valued or structured-output problems. Such settings provide a more suitable testbed for quantum kernels than scalar-valued classification

or regression, where classical kernel methods are already highly optimized and often competitive. Representative examples include multi-task learning, such as jointly predicting several related clinical outcomes (Dinuzzo, 2013); graph problems, including link prediction and molecular graph generation (Brouard et al., 2011; Shi et al., 2020); network inference and time-series prediction, for instance in climate modeling (Lim et al., 2015); and matrix-valued prediction tasks such as covariance estimation or similarity-matrix completion (Sindhwani et al., 2013; Kadri et al., 2020).

In these problems, we are faced with the task of learning a mapping between objects of different nature that each can be characterized by complex data structures (Bakir et al., 2007). The target is not an isolated scalar quantity but a structured object whose components are statistically or geometrically dependent. Therefore, designing algorithms that are sensitive enough to detect structural dependencies among these complex data is of great importance. While classical learning algorithms can be easily extended to complex inputs, more refined and sophisticated algorithms are needed to handle complex outputs. In this case, several mathematical and methodological difficulties arise and these difficulties increase with the complexity of the output space. This makes structured-output learning a substantially more challenging regime than binary/multiclass classification or scalar-valued regression. The difficulty is not merely computational; it is also representational, since the hypothesis space must encode dependencies within and across structured outputs. For this reason, structured prediction provides a natural application domain in which to investigate whether quantum operator-valued kernels can exploit richer quantum representations, including non-separable input-output interactions.

One difficulty encountered when working with structured data is that usual Euclidean methodology cannot be applied in this case. Reproducing kernels provide an elegant way to overcome this problem. Defining a suitable kernel on the structured data allows to encapsulate the structural information in a kernel function and transform the problem to a Euclidean space. Kernel-based approaches for structured output learning can be found in the literature (Tsochantaridis et al., 2005; Geurts et al., 2006; Kadri et al., 2013; Lim et al., 2015; Brouard et al., 2016; El Ahmad et al., 2024). These methods generally require an exhaustive pre-image computation (Honeine & Richard, 2011). Of special interest is supervised learning when input and output data are graph-structured. This is a complicated task that appears in various practical applications such as graph link prediction (Zhang & Chen, 2018). Graph kernels have received a lot of attention in the field of machine learning (Borgwardt et al., 2020). A few attempts have been made to introduce quantum kernels on graphs (Bai et al., 2015; Tang & Yan, 2022; Albrecht et al., 2023; Bai et al., 2024; Thabet et al., 2024).

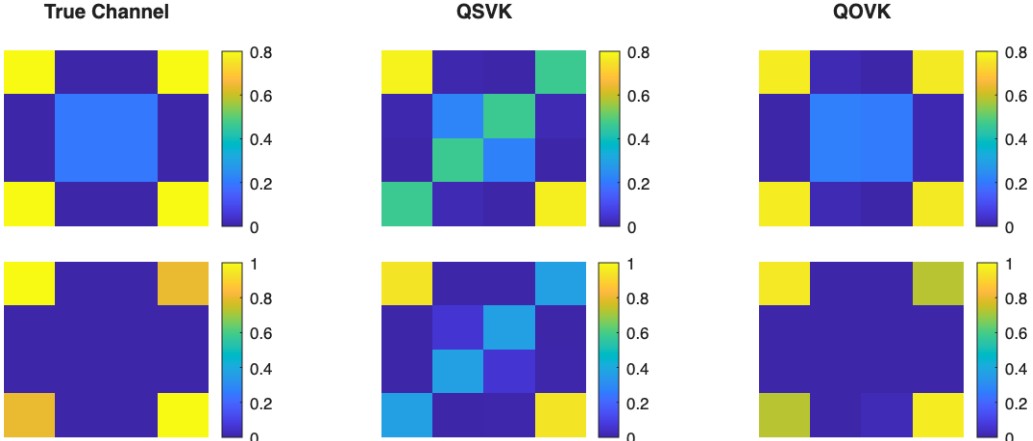

*Figure 5.* Bit-flip channel (first row) and dephasing channel (second row) estimation results with $a = 0.2$, $n = 100$ and $\alpha = 0.1$. Left plot is the Choi matrix of the correct channel from which data was created. Middle plot is the Choi matrix of the quantum channel recovered by QSVK (separable). Right plot is the Choi matrix of the quantum channel recovered by QOVK (entangled).

However, further investigations are needed to improve our understanding of: (i) How to build quantum-based structured kernels and what advantages they offer compared to classical kernels? (ii) How to solve efficiently the pre-image problem for quantum structured prediction?

## 5. Support for OVKs in QML

In this section, we provide initial support for our proposed shift towards quantum OVKs.

### 5.1. Learning with SPD Matrices: Application to Quantum Channel Estimation

As a preliminary experimental proof-of-concept, we explore how moving from quantum scalar-valued to operator-valued kernels can enhance symmetric positive-definite (SPD) matrix learning. SPD matrices are widely used in machine learning as they naturally encode covariance structures that capture relationships between variables (Minh & Murino, 2017). Here, we focus on the classical estimation of quantum channels, a task that can be formulated as an SPD matrix-valued regression problem. The objective of quantum channel estimation is to determine the dynamical transformation, known as a quantum channel, that governs the evolution of a given quantum system (Fujiwara, 2001).

In our experiment, we consider two prototypical single-qubit noise models: the bit-flip channel and the dephasing channel. Each channel depends on a parameter $a \in [0, 1]$, which specifies the probability of a bit-flip or a phase-flip. These channels provide simple yet instructive examples of quantum processes and are widely used to characterize noise in quantum devices. Details of the channels and experimental setup are provided in Appendix B. The code will be publicly available at https://gitlab.lis-lab.

fr/hachem.kadri/qovk.

We compare kernel learning performance using QSVK and QOVK. The QSVK is the fidelity kernel, defined as $k(\sigma_i, \sigma_j) = Tr(\sigma_i \sigma_j)$, while QOVK is defined by (10), with $\sigma_X^{i,j} = \sigma_i \sigma_j$. The QOVK formulation is flexible and gives rise to more expressive kernel feature spaces. The choice of $U_{YX}$ is crucial: if it is the identity, QOVK reduces to QSVK. To incorporate entanglement, we use a nonseparable $U_{YX}$, specifically the composition of a CNOT and a SWAP gate, enabling quantum correlations between input and output subsystems. Both models were trained via kernel ridge regression, with the regularization parameter selected by cross-validation.

Figure 5 illustrates an example of the true channels alongside the channels recovered using kernel ridge regression with SVKs and entangled OVKs. The OVK successfully captures the underlying structure, whereas the SVK struggles to do so. By treating the input and output spaces within a unified, entangled Hilbert space, QOVKs offer a more natural framework. They can encode matrix structure directly and model correlations between input and output spaces. Entangled QOVKs go further by using non-separable unitary operations to represent richer input-output dependencies than separable or scalar-valued kernels can capture. Further results on quantum channel recovery with QOVKs and QSVKs are reported in Appendix B.

Quantum channel estimation is considered here not merely because it is a quantum-native task, but because it provides a controlled matrix-valued learning problem in which structured input-output dependencies are explicit. Similar structural issues arise in classical settings, e.g., in covariance estimation. Thus, the experiment illustrates that QOVKs may be useful for learning problems whose outputs exhibit structure that scalar-valued kernels cannot naturally capture.

## 5.2. Quantum Implementation

An important question is whether an OVK could be implemented on a quantum computer. Here we present to our knowledge the first attempt to design a quantum circuit for OVKs. We build upon previous work on quantum implementation of scalar-valued kernels based on the swap test (Buhrman et al., 2001; Blank et al., 2020; Di Marcantonio et al., 2023). See Appendix C for more details.

We now present a quantum circuit for computing the OVK defined in (1). We consider the case where $\rho_Y$ is a pure state (i.e., $\rho_Y = |\phi\rangle_Y \langle\phi|_Y$). We generalize the quantum fidelity kernel to the operator-valued setting by introducing the quantum circuit depicted in Figure 6. See Appendix D for a detailed description of the circuit implementation. Entanglement between inputs and outputs is encoded via the matrix $U$ acting on registers $X$ and $Y$. The scalar-valued kernel can be computed via measurements of the ancillary qubit, and the separable quantum operator-valued kernel can be recovered if the operation $U$ is separable, i.e., $U$ can be written as a tensor product of unitaries acting independently on each subsystem.

The input feature matrix $\sigma_X^{x,z}$ produced by the quantum circuit is a density matrix, as it is obtained via partial measurements and is therefore Hermitian. In contrast, the general definition of an entangled QOVK does not require $\sigma_X^{x,z}$ to be Hermitian. For instance, in quantum channel estimation experiments, the QOVK is constructed from products of density matrices (i.e., $\sigma_X^{x,z} = \sigma_x\sigma_z$), which are generally non-Hermitian. The quantum circuit described here provides an initial implementation of a QOVK. It is intended to demonstrate feasibility rather than to exhaust the range of constructions enabled by the operator-valued framework. Future research should explore alternative circuit designs that can efficiently realize more general forms of $\sigma_X^{x,z}$. Other strategies could include computing the vectorization of the QOVK, which reformulates the kernel in terms of pure states rather than mixed density matrices. By encoding the kernel in terms of pure-state representations, vectorization may enable more flexible computations and expand the range of quantum features that can be incorporated, although it may require additional quantum resources. Continued research in this area is essential for implementing QOVKs that address ML tasks and explore the full potential of quantum kernels.

## 6. Alternative Views

Can research on scalar-valued kernels lead to new developments in the field of quantum kernels? Understanding the *generalization* abilities of *quantum* scalar-valued kernels could be a good alternative to explore the full power of quantum kernels. It is worth investigating generalization

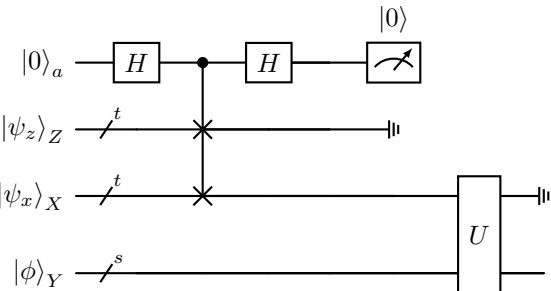

*Figure 6.* A quantum circuit for preparing a quantum state corresponding to the value of an operator-valued kernel of the form (1). The application of the partial trace on the register $Z$ by measuring the ancillary qubit in the state $|0\rangle$ produces a valid input feature matrix $\sigma^{x,z}$ for the entangled kernel.

of quantum kernel machines in both noiseless and noisy settings (Wang et al., 2021; Heyraud et al., 2022; Canatar et al., 2023). Most of the research in this topic did not incorporate recent findings on generalization of classical learning. The study of the generalization of quantum kernel machines should take into account new phenomena of modern machine learning, such as double descent and benign overfitting (Belkin et al., 2019; Bartlett et al., 2020). Only very few studies have recently appeared in the literature that attempt to look at generalization in overparameterized quantum machine learning models (Larocca et al., 2023; Peters & Schuld, 2023; Gil-Fuster et al., 2024; Thanasilp et al., 2024; Tomasi et al., 2025; Kempkes et al., 2026). Another interesting research direction is to learn the quantum feature map of quantum kernels via quantum neural networks (a.k.a. quantum parametrized circuits) (Lloyd et al., 2020; Hubregtsen et al., 2022; Gentinetta et al., 2023; Incudini et al., 2024; Glick et al., 2024; Lei et al., 2024; Rodriguez-Grasa et al., 2025). This should give rise to new quantum kernel features adapted to the task at hand. Such research avenues may also inform the design and analysis of quantum operator-valued kernels.

## 7. Conclusion

This position paper highlights quantum operator-valued kernels as a promising extension of scalar-valued quantum kernels, with the potential to exploit entanglement for richer and more expressive quantum learning models. We outline key challenges and a roadmap for advancing operator-valued kernels within QML, aiming to inspire further exploration of this framework and to foster dialogue across the machine learning and quantum computing communities.

## Acknowledgments

JT acknowledges financial support from the Program QuanTEdu-France (Grant No. ANR-22-CMAS-0001).

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

## A. Operator-Valued Kernels and Vector-Valued RKHSs

Consider the supervised learning problem where the goal is to learn a function $f : \mathcal{X} \rightarrow \mathcal{Y}$ given a training set $\{(x_i, y_i)\}_{i=1}^n$, where $x_i$ is in some space $\mathcal{X}$ and $y$ is in a Hilbert space $\mathcal{Y}$. The space $\mathcal{Y}$ can be finite or infinite-dimensional. For example, in multitask learning where the objective is to solve simultaneously $p$ learning problems, the output space $\mathcal{Y}$ can be $\mathbb{R}^p$. In functional regression, output data are curves represented by functions and the output space $\mathcal{Y}$ can be the space $L^2$ of square integrable functions. Learning the function $f$ in such situations is more challenging than finding a scalar-valued function, as is the case for standard classification or regression. The framework of scalar-valued kernels is not rich enough to learn nonlinear vector-valued functions that map complex input data to complex outputs. Operator-valued kernels provide an elegant solution to this problem. The kernel in this case is a function that takes two input data points and outputs an operator rather than a scalar as usual, i.e., $K(\cdot, \cdot) : \mathcal{X} \times \mathcal{X} \rightarrow \mathcal{L}(\mathcal{Y})$, where $\mathcal{L}(\mathcal{Y})$ is the space of bounded operators from $\mathcal{Y}$ to itself. The operator allows to encode prior information about the outputs, and then take into account the output structure. More formally,

**Definition A.1.** (psd operator-valued kernel)

A $\mathcal{L}(\mathcal{Y})$-valued kernel $K$ on $\mathcal{X} \times \mathcal{X}$ is a function $K(\cdot, \cdot) : \mathcal{X} \times \mathcal{X} \rightarrow \mathcal{L}(\mathcal{Y})$; it is positive semi-definite (psd) if:

  i. $K(\mathbf{x}, \mathbf{z}) = K(\mathbf{z}, \mathbf{x})^*$, where superscript $^*$ denotes the adjoint operator,

  ii. and, for every $n \in \mathbb{N}$ and all $\{(\mathbf{x}_i, \mathbf{y}_i)_{i=1}^n\} \in \mathcal{X} \times \mathcal{Y}$,

$$\sum_{i,j} \langle \mathbf{y}_i, K(\mathbf{x}_i, \mathbf{x}_j) \mathbf{y}_j \rangle_{\mathcal{Y}} \geq 0.$$

**Definition A.2.** (vector-valued RKHS)

A Hilbert space $\mathcal{H}$ of functions from $\mathcal{X}$ to $\mathcal{Y}$ is called a reproducing kernel Hilbert space if there is a positive semi-definite $\mathcal{L}(\mathcal{Y})$-valued kernel $K$ on $\mathcal{X} \times \mathcal{X}$ such that:

  i. $\mathbf{z} \mapsto K(\mathbf{x}, \mathbf{z})\mathbf{y}$ belongs to $\mathcal{H}$, $\forall\, \mathbf{z}, \mathbf{x} \in \mathcal{X}$, $\mathbf{y} \in \mathcal{Y}$,

  ii. $\forall f \in \mathcal{H}, x \in \mathcal{X}$, $\mathbf{y} \in \mathcal{Y}$,
$$\langle f, K(\mathbf{x}, \cdot)\mathbf{y} \rangle_{\mathcal{H}} = \langle f(\mathbf{x}), \mathbf{y} \rangle_{\mathcal{Y}} \text{ (reproducing property).}$$

A key point for learning with kernels is the ability to express functions in terms of a kernel providing the way to evaluate a function at a given point. This is possible because there exists a bijection relationship between a large class of kernels and associated reproducing kernel spaces which satisfy a regularity property. Bijection between scalar-valued kernels and RKHS was first established by Aronszajn (1950, Part I, Sections 3 and 4). Then Schwartz (1964, Chapter 5) shows that this is a particular case of a more general situation. This bijection in the case of operator-valued kernels is still valid.

**Theorem A.3.** *(bijection between vector-valued RKHS and positive semi-definite operator-valued kernel)*

*An $\mathcal{L}(\mathcal{Y})$-valued kernel $K$ on $\mathcal{X} \times \mathcal{X}$ is the reproducing kernel of some Hilbert space $\mathcal{H}$, if and only if it is positive semi-definite.*

For further reading on operator-valued kernels and their associated RKHSs, see, e.g., (Caponnetto et al., 2008; Carmeli et al., 2010; Alvarez et al., 2012; Kadri et al., 2016).

## B. Additional Experimental Details

### B.1. Experimental Setting

We consider a quantum system described by a Hilbert space $\mathcal{F}$, with the associated quantum channel represented as $\Gamma : \mathcal{S}(\mathcal{F}) \rightarrow \mathcal{S}(\mathcal{F})$, where $\mathcal{S}(\mathcal{F})$ denotes the set of density operators over $\mathcal{F}$. The quantum channel $\Gamma$ is a trace-preserving completely positive map (Watrous, 2018, Chap. 2). Quantum channel estimation can be viewed as a structured prediction problem in which both the inputs and outputs are quantum states. In our experiment, each quantum state is classically encoded as a density matrix, which is a Hermitian, positive semidefinite matrix with unit trace. Leveraging the structural properties of density matrices during learning is essential for accurately reconstructing the quantum channel from observed

*Table 1.* Experimental results on quantum channel estimation with quantum scalar-valued and quantum operator-valued kernel ridge regression. The recovery error measure is $\|\texttt{Channel}_{\text{true}} - \texttt{Channel}_{\text{learned}}\|_F$.

| Channel | QSVK | QOVK |
|---|---|---|
| Bit-flip | $0.569 \pm 0.304$ | $0.101 \pm 0.035$ |
| Dephasing | $0.525 \pm 0.297$ | $0.096 \pm 0.031$ |

data. We apply kernel-based learning algorithms to this setting, considering both quantum scalar-valued kernels (Canatar et al., 2023) and quantum operator-valued kernels as defined in (1).

We focus on learning general completely positive maps, so that the channel takes as input an SPD matrix of size $d \times d$ and outputs an SPD matrix of size $p \times p$. Now the Choi matrix representation of this channel is a matrix of size $dp \times dp$. For certain types of channels (especially ones modeling physical systems) the input and output matrices are of the same size. In our experiment we consider two prototypical single-qubit noise models: the bit-flip channel and the dephasing channel. These channels represent simple yet instructive examples of quantum processes and are widely used for characterizing noise in quantum devices. The bit-flip channel models quantum noise where a qubit is flipped from $|0\rangle$ to $|1\rangle$ and vice versa with probability $a$. It is defined, for any density matrix $\sigma$, as

$$\mathcal{E}_{\text{bf}}(\sigma) := (1 - a)\sigma + a\, X\sigma X,$$

where $X = \begin{pmatrix} 0 & 1 \\ 1 & 0 \end{pmatrix}$ is the Pauli-$X$ operator and $a \in [0, 1]$ is the bit-flip probability. The dephasing channel is a very simple decoherence process that takes a density matrix $\sigma$ to

$$\mathcal{E}_{\text{deph}}(\sigma) := (1 - a)\sigma + a\, \text{diag}(\sigma),$$

where $\text{diag}(\sigma)$ denotes the matrix comprising only the diagonal elements of $\sigma$, and $a \in [0, 1]$ measures the extent to which the off-diagonal elements, usually called coherences, are reduced in magnitude. The true channel to be estimated is fixed with a given $a$ and represented in its Choi matrix form $\Phi$, which serves as the ground truth in our evaluation. We generate $n$ training examples by sampling random density matrices $\sigma_i$, $i = 1, \dots, n$, as inputs and passing them through the channel to produce outputs $\rho_i$. To simulate noisy situations, we added a small perturbation: each output was replaced by a convex combination of the channel output and a random density matrix with mixing parameter $\alpha \in [0, 1]$, i.e.,

$$\rho_i = (1 - \alpha)\mathcal{E}_{\text{bf/deph}}(\sigma_i) + \alpha\tau_i.$$

This provides a more challenging estimation problem and tests the models' robustness to deviations from ideal data. Random density matrices are generated using QETLAB[1].

### B.2. Quantum Channel Estimation Results

The recovery errors, defined as $\|\texttt{Channel}_{\text{true}} - \texttt{Channel}_{\text{learned}}\|_F$, where $\|\cdot\|_F$ denotes the Frobenius norm, are reported in Table 1, averaged over 10 repetitions of the experiment. These results report the mean performance for channel estimation, with parameter $a$ ranging from 0 to 1 with a step size of 0.1. The OVK regression achieves significantly better performance than SVK regression.

## C. Circuit Implementation of Quantum SVKs via Swap Test

The swap test is a quantum algorithm that estimates the fidelity between two quantum states $|\psi_x\rangle$ and $|\psi_z\rangle$, i.e., $F(\psi_x, \psi_z) := |\langle\psi_x|\psi_z\rangle|^2$. Let $|\psi_x\rangle := U_x |0_t\rangle$ be an encoding of a data point $x \in \mathcal{X}$ into a quantum state of $t$ qubits obtained by applying a parametrized unitary operation $U_x$ to the initial state $|0_t\rangle := |0\rangle^{\otimes t}$. Using the density matrix formalism, the encoding corresponds to mapping an input data $x$ to a pure (i.e., rank-one) density matrix $\rho_x := U_x |0_t\rangle \langle 0_t| U_x^\dagger = |\psi_x\rangle \langle\psi_x|$. The fidelity kernel is the function $k(x, z) := \text{Tr}[\rho_x\rho_z] = |\langle\psi_x|\psi_z\rangle|^2$. The quantum circuit implementing the computation of this kernel using a swap test is given in Figure 7.

---

[1] http://www.qetlab.com.

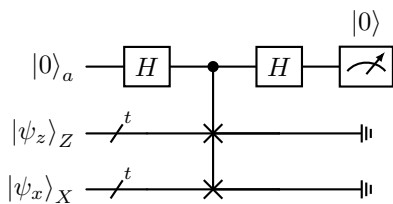

*Figure 7.* Quantum circuit for computing the fidelity kernel using a swap test. Measuring the ancillary qubit provides the fidelity between the two quantum states $|\psi_x\rangle$ and $|\psi_z\rangle$. The probability to measure the ancillary qubit in the state $|0\rangle$ is given by $\mathbb{P}(|0\rangle_a) = \frac{1}{2} + \frac{1}{2}|\langle\psi_x|\psi_z\rangle|^2$ (Schuld et al., 2015).

## D. Circuit Implementation of Quantum OVKs via Swap Test

We provide here the details of the computation of the quantum circuit implementing the QOVK given in (1) (see Figure 8). The circuit uses Hadamard and controlled-swap gates. Recall that the Hadamard gate $H$ maps a basis state $|i\rangle$, $i \in \{0, 1\}$ to the equal-weight superposition of basis states, i.e., $H|i\rangle = \frac{1}{\sqrt{2}}(|0\rangle + (-1)^i |1\rangle)$. The controlled-swap gate on two states $|\psi_x\rangle_X$ and $|\psi_z\rangle_Z$, controlled on the single qubit $|i\rangle_a$ is defined as

$$CSWAP_{aZX} |i\rangle_a |\psi_z\rangle_Z |\psi_x\rangle_X = \begin{cases} |i\rangle_a |\psi_z\rangle_Z |\psi_x\rangle_X & \text{if } i = 0, \\ |i\rangle_a |\psi_x\rangle_Z |\psi_z\rangle_X & \text{if } i = 1. \end{cases}$$

The initial state of the circuit is

$$|\Psi_0\rangle = |0\rangle_a |\psi_z\rangle_Z |\psi_x\rangle_X |\phi\rangle_Y. \tag{2}$$

Applying a Hadamard gate to the ancilla qubit of $|\Psi_0\rangle$ leads to:

$$|\Psi_1\rangle = (H_a \otimes I_{ZXY}) |\Psi_0\rangle = \frac{1}{\sqrt{2}}(|0\rangle_a + |1\rangle_a) |\psi_z\rangle_Z |\psi_x\rangle_X |\phi\rangle_Y. \tag{3}$$

$|\Psi_2\rangle$ is obtained by applying the controlled-swap gate to $|\Psi_1\rangle$:

$$|\Psi_2\rangle = CSWAP_{aZX} |\Psi_1\rangle = CSWAP_{aZX} \left(\frac{|0\rangle_a + |1\rangle_a}{\sqrt{2}}\right) |\psi_z\rangle_Z |\psi_x\rangle_X |\phi\rangle_Y$$

$$= \frac{1}{\sqrt{2}} \left( |0\rangle_a |\psi_z\rangle_Z |\psi_x\rangle_X + |1\rangle_a |\psi_x\rangle_Z |\psi_z\rangle_X \right) |\phi\rangle_Y. \tag{4}$$

A Hadamard gate is then applied to the ancillary qubit of $|\Psi_2\rangle$ giving

$$|\Psi_3\rangle = (H_a \otimes I_{ZXY}) |\Psi_2\rangle = (H_a \otimes I_{ZXY}) \frac{1}{\sqrt{2}} \left( |0\rangle_a |\psi_z\rangle_Z |\psi_x\rangle_X + |1\rangle_a |\psi_x\rangle_Z |\psi_z\rangle_X \right) |\phi\rangle_Y$$

$$= \frac{1}{2} \left[ (|0\rangle_a + |1\rangle_a) |\psi_z\rangle_Z |\psi_x\rangle_X + (|0\rangle_a - |1\rangle_a) |\psi_x\rangle_Z |\psi_z\rangle_X \right] |\phi\rangle_Y$$

$$= \frac{1}{2} \left[ |0\rangle_a (|\psi_z\rangle_Z |\psi_x\rangle_X + |\psi_x\rangle_Z |\psi_z\rangle_X) + |1\rangle_a (|\psi_z\rangle_Z |\psi_x\rangle_X - |\psi_x\rangle_Z |\psi_z\rangle_X) \right] |\phi\rangle_Y. \tag{5}$$

This pure state $|\Psi_3\rangle$ can be represented by its density matrix $\rho^{\Psi_3} = |\Psi_3\rangle \langle \Psi_3|$. Defining $\rho_x = |\psi_x\rangle \langle \psi_x|$, $\rho_z = |\psi_z\rangle \langle \psi_z|$ and omitting register subscripts for readability, $\rho^{\Psi_3}$ can be written as:

$$\rho^{\Psi_3} = \sigma^{\Psi_3} \otimes |\phi\rangle \langle \phi|, \tag{6}$$

where

$$\sigma^{\Psi_3} = \frac{1}{4} \Big[ |0\rangle \langle 0| \otimes (\rho_z \otimes \rho_x + \rho_x \otimes \rho_z + |\psi_z\rangle \langle \psi_x| \otimes |\psi_x\rangle \langle \psi_z| + |\psi_x\rangle \langle \psi_z| \otimes |\psi_z\rangle \langle \psi_x|)$$

$$+ |0\rangle \langle 1| \otimes (\rho_z \otimes \rho_x - \rho_x \otimes \rho_z - |\psi_z\rangle \langle \psi_x| \otimes |\psi_x\rangle \langle \psi_z| + |\psi_x\rangle \langle \psi_z| \otimes |\psi_z\rangle \langle \psi_x|)$$

$$+ |1\rangle \langle 0| \otimes (\rho_z \otimes \rho_x - \rho_x \otimes \rho_z + |\psi_z\rangle \langle \psi_x| \otimes |\psi_x\rangle \langle \psi_z| - |\psi_x\rangle \langle \psi_z| \otimes |\psi_z\rangle \langle \psi_x|)$$

$$+ |1\rangle \langle 1| \otimes (\rho_z \otimes \rho_x + \rho_x \otimes \rho_z - |\psi_z\rangle \langle \psi_x| \otimes |\psi_x\rangle \langle \psi_z| - |\psi_x\rangle \langle \psi_z| \otimes |\psi_z\rangle \langle \psi_x|) \Big].$$

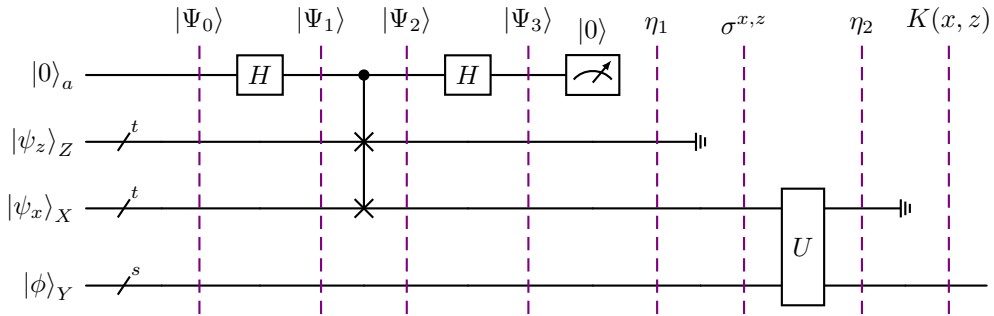

*Figure 8.* A quantum circuit implementation of the operator-valued kernel given in (1).

The state $\eta_1$ is then obtained by measuring the ancilla qubit in the state $|0\rangle$, i.e.,

$$
\begin{aligned}
\eta_1 &= \frac{\mathrm{Tr}_a[\rho^{\Psi_3}(|0\rangle\langle 0|_a \otimes I_{ZXY})]}{\mathrm{Tr}[\rho^{\Psi_3}(|0\rangle\langle 0|_a \otimes I_{ZXY})]} \\
&= \frac{\rho_z \otimes \rho_x + \rho_x \otimes \rho_z + |\psi_z\rangle\langle\psi_x| \otimes |\psi_x\rangle\langle\psi_z| + |\psi_x\rangle\langle\psi_z| \otimes |\psi_z\rangle\langle\psi_x|}{2\left(1 + |\langle\psi_z|\psi_x\rangle|^2\right)} \otimes |\phi\rangle\langle\phi|.
\end{aligned}
\tag{7}
$$

The input feature matrix $\sigma_X^{x,z}$ is obtained in the register $X$ after measuring the subsystem $Z$ by partial tracing $\eta_1$:

$$
\sigma_X^{x,z} = \frac{\rho_x + \rho_z + \langle\psi_x|\psi_z\rangle |\psi_x\rangle\langle\psi_z| + \langle\psi_z|\psi_x\rangle |\psi_z\rangle\langle\psi_x|}{2\left(1 + |\langle\psi_z|\psi_x\rangle|^2\right)}.
\tag{8}
$$

Once we have prepared the feature matrix (8), we can proceed with the evaluation of the operator-valued kernel. First, entanglement between $X$ and $Y$ is introduced by a non-separable unitary $U$, giving the state $\eta_2$:

$$
\eta_2 = U_{YX}(|\phi\rangle\langle\phi|_Y \otimes \sigma_X^{x,z})U_{YX}^\dagger.
\tag{9}
$$

The evaluation of the operator-valued kernel is then obtained by measuring the register $X$ of $\eta_2$:

$$
K(x,z) = \mathrm{Tr}_X[U_{YX}(|\phi\rangle\langle\phi|_Y \otimes \sigma_X^{x,z})U_{YX}^\dagger].
\tag{10}
$$

