# OpenReview forum: "Position: Quantum Kernel Machines Should Move Beyond Scalar-Valued Kernels to Realize Their Potential"
_ICML.cc/2026/Position_Paper_Track — ICML 2026 Position Paper Track regular_

### Official Review · Reviewer_CEHw · 2026-02-15

**Significance:** 2
**Argument Clarity:** 2
**Rating:** 4
**Confidence:** 3

**Questions:**

N/A

**Alternative Views Section:**

Yes

**Compliance With Llm Reviewing Policy A Conservative:**

Affirmed.

**Discussion Potential:**

2

**Final Justification:**

After reading the rebuttal, my concerns have been adequately addressed, as the authors provided a clear and comprehensive response that clarifies both the complexity of the discussion and the broader relevance of the proposed framework. Therefore, I support acceptance of the paper.

**Paper Summary:**

This paper argues that progress in QML requires moving beyond scalar-valued kernels toward more expressive kernel frameworks and presents an initial proof-of-concept illustrating how quantum entangled operator-valued kernel formulations can reveal structural dependencies that remain difficult to access for scalar-valued kernel methods.

**Position:**

Yes

**Position In Title:**

Yes

**Related Work:**

2

**Strengths And Weaknesses:**

Strengths

1. The paper provides a clear mathematical definition and possible circuit designs for QOVKs.

2. The paper is well written with fluent logic flow.

Weaknesses

1. Lack of possible complexity analysis. The paper acknowledges that "This suggests that quantum algorithms could have a more substantial impact in the operator-valued setting, although the implementation of quantum solvers on current quantum devices still faces many challenges."

(1) Possible challenges and complexity analysis should be provided in the paper to support the position better.

(2) Discussions on how QOVKs can reduce the challenges and complexity in computation should be included to support the authors' claim.

2. Quantum Channel Estimation is a quantum-native problem. Whether the proposed QOVKs can be extended to more general problems should be discussed in the paper.

**Support:**

2

---

> ### Author Rebuttal · Authors · 2026-03-30
>
> We thank the reviewer for the useful comments.
>
> ### **Complexity analysis**
>
> [See also our reply for reviewer JRdB]
>
> We agree that this should be made more explicit. The paper already states the main point in Section 3: in the scalar-valued case, learning uses an $n\times n$ kernel matrix, whereas in the operator-valued case it uses a block matrix of size $np\times np$, which is significantly more expensive to manipulate. We will add a short quantitative paragraph: classically, dense training scales from $O(n^3)$ time and $O(n^2)$ memory in the scalar case to $O((np)^3)$  time and $O(n^2p^2)$ memory in the operator-valued case. On the quantum side, the relevant primitives are Grover/amplitude amplification for search-type subroutines with quadratic improvements (Roget et al., 2022) and quantum linear-system solvers, whose dependence on matrix dimension is polylogarithmic under the usual sparse-access or block-encoding assumptions (Morales et al., 2024).
>
> We will also make the challenges explicit: state preparation, oracle/block-encoding access to the kernel matrix, dependence on condition number and precision, measurement/readout overhead, and hardware depth/noise. Importantly, QOVKs do not reduce the raw classical asymptotic size of the problem; rather, they make quantum acceleration more meaningful because the classical bottleneck is larger. In addition, they reduce a modeling bottleneck by avoiding repeated scalarization of structured outputs and by allowing richer non-separable input-output couplings. This is already the motivation behind our discussion of entangled QOVKs and Action 2.
>
> ### **From quantum-native channel estimation to broader structured-output learning**
>
> [See also our reply for reviewer mEPs]
>
> OVKs have already been successfully applied in a wide range of classical ML problems, including multi-task learning, structured-output prediction, network inference, multiview learning, operator/functional regression, PDE learning, and graph prediction (see Section 3 and references therein). These examples demonstrate that the OVK framework is well suited to problems with structured or multi-dimensional outputs.
>
> The quantum channel estimation experiment in Section 5.1 should therefore be viewed as a first proof-of-concept, rather than a limitation. It is particularly well aligned with the OVK framework because it is a matrix-valued regression problem with strong structural constraints, making it a natural testbed for OVKs.
>
> Importantly, the advantages suggested by QOVKs are not specific to quantum-native tasks, but rather to structured-output learning more broadly. For instance, applications such as learning covariance matrices for brain-computer interfaces or positive semidefinite (PSD) similarity matrix completion naturally involve structured matrix outputs and are well suited to the OVK framework. Such problems have been addressed recently using classical and quantum-inspired ML approaches  (Huang et al. 2017, Kadri et al., 2020). QOVKs can be viewed as a nonlinear and quantum alternative of such methods.
>
> Moreover, classical non-separable OVKs have been shown to outperform separable ones in applications such as supervised dimensionality reduction and multi-output regression (Huusari et al., 2021). A quantum implementation provides a pathway to realize such expressive kernels in high-dimensional feature spaces with the potential for improved computational efficiency.
>
> We will revise the paper to make this broader perspective more explicit and to better connect the proposed framework to existing applications of operator-valued kernels beyond quantum-native settings.

---

> > ### Author Rebuttal · Reviewer_CEHw · 2026-04-01
> >
> > I appreciate the authors’ comprehensive response.

---

### Official Review · Reviewer_jRdB · 2026-03-09

**Significance:** 4
**Argument Clarity:** 3
**Rating:** 4
**Confidence:** 4

**Questions:**

1. Could you please elaborate on the necessity of the claimed advantage of QOVKs over OVKs in matrix operations? To the best of my knowledge, in kernel-based machine learning, situations where operator-valued kernel matrix computations are required may not be very common.

**Alternative Views Section:**

Yes

**Compliance With Llm Reviewing Policy A Conservative:**

Affirmed.

**Discussion Potential:**

4

**Paper Summary:**

This paper proposes a future research direction for quantum kernel machines, advocating the extension of quantum kernels from scalar-valued to operator-valued forms, a setting that has not yet been explored in the current literature. In the case of scalar-valued kernels and relatively simple learning tasks, classical kernel methods have already been extensively studied, enabling fast computation and efficient representation. As a result, the potential advantages of quantum kernels are difficult to demonstrate in such settings. The authors argue that when quantum kernels are extended to the operator-valued setting, their representational potential can be more fully exploited. To this end, the paper introduces the definition of Entangled Quantum Operator-Valued Kernels (QOVKs) and initiates the study of QOVKs. The authors further outline a set of actionable steps for advancing this research direction and provide practical evidence demonstrating the potential advantages of QOVKs as well as the feasibility of their concrete implementation.

**Position:**

Yes

**Position In Title:**

Yes

**Related Work:**

4

**Strengths And Weaknesses:**

Strengths：
1. The paper presents a thorough survey and discusses the current limitations of quantum scalar-valued kernels (QSVKs) from multiple perspectives, thereby motivating the necessity of introducing quantum operator-valued kernels (QOVKs). This perspective has the potential to stimulate future research in this area.
2. This paper introduce a specific class of QOVKs, namely entangled QOVKs, thereby providing a foundational and guiding research framework for this emerging area.
3. The authors conduct experimental validation of QOVKs and implement them on a quantum computer, further demonstrating the research value and practical feasibility of the proposed direction.

Weakness:
1. Regarding the authors’ statement that “We strongly believe that we have to focus on more complicated ML tasks where classical kernel methods face clear limitations in order to reveal the full potential of quantum kernels,” the manuscript lacks sufficient verbal argumentation. (Although quantum channel estimation is mentioned as a proof, it may not necessarily fall under the category of “complicated ML tasks.”)

**Support:**

2

---

> ### Author Rebuttal · Authors · 2026-03-30
>
> We thank the reviewer for the useful comments.
>
> ### **Complicated ML tasks**
>
> [See also our reply for reviewer mEPs]
>
> We agree that complicated ML tasks should be made more clearer. By “complicated ML tasks” we mean tasks that are not scalar-valued, but instead involve vector-valued, matrix-valued, or structured-valued outputs, together with nontrivial dependencies across outputs and richer input-output couplings. This is exactly the regime in which OVKs are useful classically, and why the paper points to multitask learning, operator/functional regression, multiview learning, PDE learning, network inference, and graph-structured prediction as motivating examples.  The quantum channel estimation experiment follows this line: it is a matrix-valued prediction problem and serves as a proof-of-concept illustrating the benefit of moving beyond scalar-valued kernels.
>
> This argument is already present in Sections 2 and 4 but can be made clearer. The broader motivating class of “complicated tasks” is the one emphasized in Action 4: structured prediction problems such as graph-structured prediction and network inference. So we will make this more clearer in the paper by
> * stating explicitly that: “more complicated ML tasks” refers to vector-, matrix-, and structured-output learning problems;
> * expanding Action 4 with representative structured output prediction applications already studied with classical OVKs, including graph link prediction (Brouard et al., 2011), network inference (Lim et al., 2015), operator regression (Kadri et al., 2016) and PDE learning (Stepaniants, 2023). This will make clear that the suitability of the operator-valued framework for such applications is already supported by the classical literature, which in turn motivates the study of its quantum counterpart.
>
> ### **OVK matrix computations**
> > Could you please elaborate on the necessity of the claimed advantage of QOVKs over OVKs in matrix operations? To the best of my knowledge, in kernel-based machine learning, situations where operator-valued kernel matrix computations are required may not be very common.
>
> Thank you for raising this important question. We agree that the role of OVK matrix computations should be clarified, and we will make this more explicit in the revised version.
>
> OVK matrix computations are not marginal and are used in many ML tasks: they arise precisely in the settings that motivate OVKs, including multi-task learning, functional/operator regression, network inference, PDE learning, and structured output prediction. In these problems, the kernel matrix is no longer an $n \times n$ scalar Gram matrix, but a block matrix of size $np \times np$, where $p$ is the output dimension. As noted in Section 3, this significantly increases the computational burden, as matrix inversion, eigendecomposition, and optimization scale with the full $np \times np$ matrix, especially when both $n$ and $p$ are large.
>
> From a computational perspective, this is precisely the regime where quantum methods may have more room to provide benefits. In the scalar-valued setting, Classical kernel methods typically require $O(n^2)$ memory and up to $O(n^3)$ computational time. In the operator-valued case, these costs scale to $O(n^2 p^2)$ memory and $O((np)^3)$ time in the worst case. For decomposable/separable OVKs, Kronecker or Sylvester structure can reduce this burden substantially, but this computational simplification is also a modeling restriction: it is exactly the structure that more expressive non-separable / entangled QOVKs aim to move beyond.
>
> On the quantum side, two main classes of routines are relevant. First, quantum linear system solvers such as HHL (Harrow et al., 2009) and more recent block-encoding / QSVT approaches (Gilyén et al., 2019) can solve linear systems in time $\tilde{O}(\kappa \log N / \epsilon)$, where N is the matrix dimension (Morales et al., 2024). Second, Grover search / amplitude amplification can yield quadratic improvements in search or sampling subroutines (Grover, 1996, Brassard et al., 2002).
>
> When moving from scalar-valued kernels ($N=n$) to operator-valued kernels ($N=np$), the classical computational cost increases polynomially in $np$. In contrast, quantum routines can exhibit more favorable scaling: quantum linear system solvers retain polylogarithmic dependence on the matrix dimension $N$, while amplitude amplification-type methods reduce the complexity from $O(N/\gamma^2)$ to $O(\sqrt{N}/\gamma)$ (Roget et al., 2022).
>
> This does not by itself establish a quantum advantage for QOVKs, but it highlights why the operator-valued setting is structurally more favorable for potential computational gains. We will clarify this point in the revised paper.
>
> Finally, we note that these quantum speedups are conditional on standard assumptions (e.g., efficient state preparation and quantum access to data) and typically return the solution in quantum form.

---

> > ### Author Rebuttal · Reviewer_jRdB · 2026-04-06
> >
> > I appreciate the authors’ comprehensive response.

---

### Official Review · Reviewer_2i2C · 2026-03-12

**Significance:** 1
**Argument Clarity:** 3
**Rating:** 4
**Confidence:** 1

**Questions:**

none

**Alternative Views Section:**

Yes

**Compliance With Llm Reviewing Policy A Conservative:**

Affirmed.

**Discussion Potential:**

2

**Final Justification:**

I appreciate the authors’ response, which covers the major raised concerns. I have increased my score accordingly.

**Paper Summary:**

At the end, there is no clear call for actions. The paper seems more like a comparative analysis of different thresholding and score scaling techniques.

**Position:**

Yes

**Position In Title:**

Yes

**Related Work:**

2

**Strengths And Weaknesses:**

Strengths:
The paper focuses on the stated position, aiming to demonstrate the relevance of moving beyond scalar-valued kernels to operator-valued ones.
This topic is relevant to the ICML community, considering the recent interest in quantum machine learning.

Weaknesses:
The stated position is not supported by any experimental analysis. The paper would have benefitted from some experimental analysis with empirical comparative analysis to demonstrate the relevance and importance of the stated position. It would have been relevant to consider several variants of quantum kernels, in scalar-valued and operator-valued frameworks.
The alternative views section does not provide in-depth analysis.

**Support:**

1

---

> ### Author Rebuttal · Authors · 2026-03-30
>
> We thank the reviewer for the comments. We address each point below.
>
> ### **Call for actions**
>
> We respectfully disagree that the paper lacks a clear call for action. Section 4 is explicitly titled “Call to Action” and proposes four concrete research directions: (1) quantum implementation of OVKs, (2) quantum entangled OVKs, (3) a C*-algebraic extension, and (4) application to quantum structured prediction.
>
> ### **Thresholding and score scaling techniques**
>
> We believe this remark may stem from a misunderstanding as the paper does not study thresholding methods or score scaling techniques.
>
> ### **Experimental support**
>
> We also respectfully disagree with the statement that the position is unsupported experimentally. The paper contains an explicit section, “Support for OVKs in QML” (Section 5), which provides initial support for the proposed position. In Section 5.1, we report an experimental comparison between a scalar-valued quantum kernel  and an entangled QOVK on quantum channel estimation, formulated as an SPD matrix-valued regression problem. As this is a position paper, our aim is not to provide an exhaustive benchmark over many datasets and kernels, but to articulate and support a research direction with an initial proof-of-concept.
>
> ### **Alternative view**
>
> The goal of Section 6 is not to provide a second full position, but to acknowledge other legitimate research paths including continued work on scalar-valued quantum kernels.

---

> > ### Author Rebuttal · Reviewer_2i2C · 2026-04-04
> >
> > I appreciate the authors’ response, which covers the major raised concerns. I have increased my score accordingly.

---

### Official Review · Reviewer_mEPs · 2026-03-16

**Significance:** 3
**Argument Clarity:** 3
**Rating:** 5
**Confidence:** 3

**Questions:**

N/A

**Alternative Views Section:**

Yes

**Compliance With Llm Reviewing Policy A Conservative:**

Affirmed.

**Discussion Potential:**

3

**Paper Summary:**

In this position paper, the authors argue that prior work on quantum kernels has focused too heavily on scalar-valued kernels, which are unlikely to demonstrate clear advantages over classical kernels on standard classification and regression tasks. They propose that quantum operator-valued kernels (QOVKs) offer a richer and more expressive framework that is better suited to complex structured prediction problems.

**Position:**

Yes

**Position In Title:**

Yes

**Related Work:**

3

**Strengths And Weaknesses:**

Strength:
1. The paper addresses an important pain point in current research on quantum kernel methods: the difficulty of demonstrating clear advantages over classical kernel functions. It provides a useful perspective on where the potential strengths of quantum kernel methods may lie and motivates a shift toward more challenging quantum structured prediction, where richer kernel formulations could offer meaningful benefits.


2. A thorough discussion is provided on the current challenges facing quantum kernels. In particular, the paper highlights that the expressive power of quantum models may actually hinder generalization. Moreover, designing suitable quantum kernels is not straightforward, as kernel evaluation may require exponentially many measurements. In addition, when using a large number of qubits, the resulting kernel matrix can become close to the identity matrix, leading to overfitting and poor generalization performance. And then demonstrate the potential of new proposed QOVKs can be served as solutions.

3. Preliminary experiment on single-qubit quantum channels estimation where the reconstructed Choi matrices produced by QOVK more closely match the ground-truth channels than those produced by QSVK. And demonstrate the potential design of its quantum circuit.



Weakness:
1. Although the proposition is insightful and provide several potential directions to further investigate the results, most practical machine learning problems still involve standard classification and regression tasks. Providing concrete real-world applications where complex structured prediction with quantum kernels is necessary would strengthen the argument. In particular, examples demonstrating the relevance of quantum operator-valued kernels for realistic structured prediction problems could help clarify the practical impact of the proposed framework.

**Support:**

3

---

> ### Author Rebuttal · Authors · 2026-03-30
>
> We thank the reviewer for his constructive comments.
>
> ### **Relevance of quantum operator-valued kernels for realistic structured prediction problems**
>
> The paper points to real-world applications, especially in structured prediction, where quantum OVKs could be relevant, primarily in Action 4 of Section 4 but also in Section 3 and the introduction. We agree that highlighting them more clearly would strengthen the presentation.
>
> Our goal in this position paper is not to claim that the practical impact of QOVKs has already been established. Rather, our point is that many important real-world learning problems are naturally vector-valued, matrix-valued, or structured-output problems, and that these settings provide a more appropriate regime in which to investigate the potential of quantum kernels than scalar-valued prediction alone.
>
> Representative examples include multi-task learning (e.g., jointly predicting multiple related clinical outcomes [Dinuzzo, 2013]), graph prediction (e.g., molecular property prediction or link prediction [Brouard et al., 2011]), network inference and multivariate time-series prediction (e.g., modeling and predicting climate variable [Lim et al., 2015]), and matrix-valued prediction (e.g., covariance estimation or similarity matrix completion [Sindhwani et al., 2012; Kadri et al., 2020]). These are precisely the types of problems for which  SVKs become restrictive and for which OVKs provide a natural framework. This is also consistent with our proof-of-concept in Section 5.1: quantum channel estimation is not a scalar prediction task, but a matrix-valued regression problem, and was included precisely as an initial illustration of the kind of setting where QOVKs may reveal advantages over SVKs.
>
> More generally, when we refer to “standard classification and regression,” we mean scalar-valued classification and regression, as stated in Section 2.4. Our point is not that these tasks are unimportant, but that they are already very well served by classical methods, which makes them a limited testbed for assessing the full potential of quantum kernels.
>
> To address the reviewer’s suggestion, we will revise the paper to make this point more explicit. In particular, we will clarify in the Introduction and Section 2.4 that “standard classification/regression” refers to scalar-valued prediction, and we will expand Action 4 with representative structured output prediction applications that have already been studied with classical OVKs, including those mentioned above. This will make clear that the suitability of the operator-valued framework for such applications is already supported by the classical literature, which in turn motivates the study of its quantum counterpart.
>
> #### **Additional references**
>
> Brouard, C., d'Alché-Buc, F., & Szafranski, M. (2011). Semi-supervised penalized output kernel regression for link prediction. ICML.
>
> Dinuzzo, F. (2013). Learning output kernels for multi-task problems. Neurocomputing, 118, 119-126.
>
> Sindhwani, V., Quang, M. H., & Lozano, A. C. (2013). Scalable matrix-valued kernel learning for high-dimensional nonlinear multivariate regression and granger causality. UAI.

---

> > ### Author Rebuttal · Reviewer_mEPs · 2026-04-02
> >
> > Thanks for the reviewer's reply.

---

### Decision · Program_Chairs · 2026-04-30

**Decision:**

Accept (regular)

**Comment:**

This paper makes a strong position-paper contribution. It identifies a real bottleneck in quantum kernel research: most existing work tests quantum kernels in the scalar-valued setting on standard prediction tasks where classical kernel methods are already mature and effective, making it difficult to observe meaningful gains.

The proposed shift toward operator-valued and even C*-algebraic kernels is conceptually well motivated, technically informed, and connected to structured prediction problems where richer input-output couplings matter more. The paper also goes beyond pure advocacy by introducing a concrete class of entangled QOVKs, providing a proof-of-concept experiment on quantum channel estimation, and outlining an initial quantum-circuit implementation.

The score summary is 5, 4, 4, 4: one accept and three borderline accepts. Reviewers find the core direction important, the technical discussion thorough, and the proof-of-concept sufficient for a position paper. The main reservations are moderate rather than fundamental: reviewers ask for clearer discussion of realistic structured-output applications, more explicit complexity analysis, and stronger articulation of how QOVKs extend beyond the quantum-native channel-estimation example. These concerns are addressed constructively in rebuttal, and no reviewer remains strongly negative.

The recommendation is accept. The paper presents a clear thesis, engages a credible alternative, and offers enough conceptual and technical support to justify the proposed research shift. It does not claim that QOVKs are already a proven solution; rather, it argues persuasively that they define a richer and more appropriate regime in which to test the promise of quantum kernels.